# Urinary Tract Infections Caused by *Klebsiella pneumoniae* and Prolonged Treatment with Trimethoprim/Sulfamethoxazole

**DOI:** 10.3390/microorganisms13020422

**Published:** 2025-02-14

**Authors:** Rumen Filev, Mila Lyubomirova, Boris Bogov, Alexander Kolevski, Victoria Pencheva, Krasimir Kalinov, Lionel Rostaing

**Affiliations:** 1Department of Nephrology, Internal Disease Clinic, University Hospital “Saint Anna”, 1750 Sofia, Bulgaria; rsfilev@gmail.com (R.F.); mljubomirova@yahoo.com (M.L.); bbogov@yahoo.com (B.B.); 2Faculty of Medicine, Medical University Sofia, 1504 Sofia, Bulgaria; 3Department of Microbiology, University Hospital “Alexandrovska”, 1431 Sofia, Bulgaria; al.kolevski@gmail.com (A.K.); penchevvictoria@gmail.com (V.P.); 4Comac-Medical Ltd., 1404 Sofia, Bulgaria; krassimir.kalinov@comac-medical.com; 5Nephrology, Hemodialysis, Apheresis and Kidney Transplantation Department, Grenoble University Hospital, 38043 Grenoble, France; 6Internal Disease Department, Grenoble Alpes University, 38043 Grenoble, France

**Keywords:** urinary tract, infection, *Klebsiella pneumonia*, trimethoprim/sulfamethoxazole

## Abstract

Urinary tract infections (UTIs) are among the most prevalent bacterial infections, with *Klebsiella pneumoniae* emerging as a significant multidrug-resistant pathogen, particularly in healthcare settings. The frequent utilization of β-lactams and fluoroquinolones has contributed to the reduced clinical application of trimethoprim-sulfamethoxazole (TRS). Nevertheless, this reduced use may have preserved its efficacy as a second-line therapy. The aim of this study is to determine whether TRS can be a cost-effective long-term solution for patients with multidrug-resistant *K. pneumoniae* UTIs. This study evaluated the effectiveness of a structured, long-term TRS regimen in 11 patients with recurrent *K. pneumoniae* UTIs who had failed at least two prior antibiotic treatments. Patients were carefully selected, and the collected data were analyzed using descriptive analysis. The findings demonstrated microbiological eradication and symptomatic relief in all patients, with only one case of a delayed allergic reaction. All patients had a negative urine microbiology analysis after at least two unsuccessful treatment attempts over a period of 6 to 18 months. These results underscore TRS’s potential as a cost-effective and efficient second-line treatment, particularly in settings with limited therapeutic options. Its affordability, manageable side effect profile, and ability to target biofilm-associated infections further highlight its therapeutic value.

## 1. Introduction

Urinary tract infections (UTIs) are among the most common bacterial infections, affecting millions of individuals globally each year [1]. These infections occur when pathogenic microorganisms, predominantly bacteria, colonize the urinary tract, leading to symptoms such as dysuria, pollakiuria, and lower abdominal discomfort [2]. Over the last decades, *Klebsiella pneumoniae* has emerged as a significant pathogen, particularly in healthcare environments and among immunocompromised individuals [3]. *K. pneumoniae* is a Gram-negative bacterium that has emerged as a major pathogen responsible for both community-acquired and nosocomial UTIs, particularly in immunocompromised individuals and those with chronic underlying conditions [4]. Its ability to form biofilms, evade immune responses, and acquire resistance through horizontal gene transfer has made it a significant challenge in clinical practice [5].

The treatment of *K. pneumoniae* UTIs has become increasingly difficult due to the bacterium’s resistance to many first-line antibiotics, including fluoroquinolones and β-lactams [6]. The emergence of extended-spectrum β-lactamase (ESBL)-producing strains, which confer resistance to third-generation cephalosporins, has further restricted available therapeutic options [4,5]. A growing body of research highlights the rising prevalence of multidrug-resistant (MDR) *K. pneumoniae*, which is resistant to at least three major antibiotic classes. Of particular concern is the increasing resistance to carbapenems, a last-resort antibiotic, driven by the production of carbapenemase enzymes such as KPC (*K. pneumoniae* carbapenemase) [7].

Trimethoprim-sulfamethoxazole (TRS) was historically used as a second-line treatment for UTIs. However, its clinical use has declined over time due to rising resistance rates, which have been reported to range from 50% to over 70%, depending on the region and patient population [4,7]. Despite this, TRS remains a valuable option in resource-limited settings or for patients with few alternative treatments, particularly when administered in structured regimens tailored to individual factors such as renal function [5]. However, the frequent use of β-lactams and fluoroquinolones in treating *K. pneumoniae* urinary infections has led to a decline in the clinical use of TRS [8]. It can therefore be hypothesized that TRS has retained its effectiveness, as resistance to it has been less strongly selected for over time.

The clinical application of TRS presents several challenges. While evidence suggests that it is effective in treating certain carbapenem-resistant *Enterobacterales* (CRE) infections, the emergence of resistance during therapy remains a major limitation. For instance, studies conducted by the CRACKLE-1 consortium found that only 29% of CRE isolates were initially susceptible to TRS, with a significant proportion developing resistance over time [7]. Moreover, resistance mechanisms in *K. pneumoniae*, such as efflux pumps, biofilm formation, and enzyme production, further complicate the management of recurrent UTIs and highlight the need for innovative treatment approaches [7].

In addition to TRS, international guidelines recommend novel agents such as ceftazidime-avibactam, meropenem-vaborbactam, and plazomicin for the treatment of MDR *K. pneumoniae* infections [7]. However, the high costs and limited availability of these agents restrict their widespread use in many regions.

As a result, balancing affordability, accessibility, and efficacy is crucial in managing MDR *K. pneumoniae* UTIs. TRS remains an alternative of significant interest due to its low cost, manageable side-effect profile, and potential for use in combination regimens [7].

This study aims to address key gaps in the management of recurrent UTIs caused by MDR *K. pneumoniae* by evaluating the long-term efficacy of a structured TRS regimen. Leveraging recent insights into resistance patterns, biofilm formation, and emerging therapeutic strategies, this research seeks to provide a comprehensive understanding of TRS’s role in the evolving landscape of antimicrobial resistance.

The aim is to demonstrate whether TRS can be used as a long-term, effective and cost-effective second-line therapy for patients with persistent MDR *K. pneumoniae* infections.

## 2. Materials and Methods

### 2.1. Study Design

This study design was approved by the Institutional Review Board (approval code: 01/02-D/24, date: 1 February 2024). All patients were recruited during hospitalization in the nephrology clinic at Saint Anna Hospital. They were eligible for inclusion if they had a confirmed *K. pneumoniae* UTI with at least two unsuccessful antibiotic treatments and persistent symptoms between treatment attempts. An ultrasound examination was required to confirm a lower urinary tract infection, showing bladder wall thickening greater than 5 mm, post-void residual volume, echogenic debris in the bladder, or bladder mucosal irregularity. Patients with negative microbiology results for more than six months were excluded.

Microbiological analysis had to indicate susceptibility to TRS. Patients with known allergies to antibiotics or other medications were not included, although one case of a delayed reaction was observed despite an initial absence of allergy history. Female patients were required to have a negative pregnancy test before inclusion. All participants provided informed consent for long-term follow-up, and only individuals over 18 years old were included in this study.

This study included 18 patients with a history of at least two unsuccessful antibiotic treatments of a UTI caused by *K. pneumonia*. Demographic and clinical data were collected for all participants, including comorbidities, kidney function, and prior treatment regimens. Patient data were analyzed, taking into account demographics, kidney function—assessed by serum creatinine and the estimated glomerular filtration rate (eGFR) using CKD-EPI 2021—as well as additional factors such as chronic kidney disease (CKD), diabetes, and hypertension.

The cohort was divided into two groups: the first group comprised 11 patients (seven females and four males), and microbiological analysis revealed that all of them exhibited a positive response to treatment with TRS. The second group, comprising seven patients (five females and two males), served as the control group. All patients in the control group exhibited positive results from microbiology urine analysis for *K. pneumoniae*, yet all demonstrated resistance to TRS treatment according to the antibiogram.

### 2.2. Sample Collection

Urine samples were obtained from all patients under strict aseptic conditions using a midstream clean-catch technique. Prior to collection, patients were instructed to cleanse the genital area thoroughly with an antiseptic wipe to reduce the risk of contamination. In instances where catheterization was required, urine samples were collected directly from the catheter port using a sterile syringe. All samples were promptly transferred into sterile containers to prevent contamination. Following collection, samples were immediately transported to the microbiology laboratory. Upon arrival, they were processed within one hour. A calibrated loop, delivering 10 µL of urine, was used to inoculate MacConkey and blood agar plates. The plates were incubated at 37 °C for 18–24 h under aerobic conditions. Significant bacterial growth was considered to be present when colony-forming units (CFUs) exceeded 10^5^ per mL of urine. The microbiological examination of urine for all patients confirmed significant growth of *K. pneumoniae*—greater than 10^5^. All uroinfections were symptomatic.

For all patients, after obtaining the antibiogram, a suitable first-line antibiotic (administered intravenously) was selected for 7–10 days. Following the successful completion of this initial therapeutic course and hospitalization discharge, TRS (800/160 mg twice daily) therapy was prescribed with adjustments made based on renal function and according to a structured schedule designed to optimize therapeutic efficacy while minimizing potential adverse effects. The long-term second-line treatment regimen with TRS was carried out in six sequential steps, as follows:Treatment with 2 × 2 tablets—Administered daily for 5 days post-hospital discharge.Treatment with 2 × 2 tablets—Administered every other day for 2 weeks.Treatment with 2 × 2 tablets—Administered every two days for 2 weeks.Treatment with 2 × 2 tablets—Administered every three days for 2 weeks.Treatment with 2 × 2 tablets—Administered every four days for 2 weeks.Treatment with 2 × 2 tablets—Administered every five days for 2 weeks.

For patients with eGFR (CKD-EPI 2021) values between 15 and 30 mL/min/1.73 m^2^, the dosage was adjusted to 2 × 1 tablets following the same schedule.

A urine microbiology and sediment test was required every two weeks following hospital discharge, throughout the duration of the treatment. This interval was chosen because the microbiologists in our team wished to closely monitor the patients when the dosage of TRS was being adjusted.

### 2.3. Quality Control

To ensure the accuracy and reliability of results, quality control measures were implemented throughout the microbiological testing process. Laboratory equipment was regularly calibrated, and all culture media were prepared and validated in accordance with established protocols.

### 2.4. Statistical Methods

This study is exploratory in nature. Only descriptive methods were applied.

Methods for data summarizing used were as follows:-Categorical parameters were analyzed by absolute and relative (percentage) frequencies.-Continuous parameters were analyzed by the number of measurements, arithmetic mean, standard deviation (SD), minimum, 1st quartile, median, 3rd quartile, and maximum values.

SAS^®^ package ver. 9.4, was used to produce summary tables.

## 3. Results

Eleven patients were included, of whom seven were female (63.6%) and four were male (36.4%), reflecting a predominance of females in the patient cohort. The participants varied in age and medical backgrounds (see Table 1 and Table 2) and had previously been treated with different antibiotics. All patients presented to the nephrology clinic with a positive urine culture for *K. pneumoniae*, identified as a multidrug-resistant strain based on antibiogram results. Each patient had a documented history of at least two unsuccessful antibiotic treatments for UTIs associated with *K. pneumoniae* prior to inclusion in this study, with a duration of 6 to 18 months.

A significant proportion of patients had comorbidities such as diabetes, hypertension, and chronic kidney disease (CKD) at varying stages. In examining the cohort for chronic kidney disease (CKD), diabetes, and hypertension, distinct patterns emerged. CKD was observed across multiple stages, although some patients had no history of the condition. Stage III CKD was the most prevalent, affecting four patients (Grade IIIa and IIIb, each present in two patients), indicating a moderate decline in kidney function. One patient was classified as stage I, reflecting early-stage kidney disease, while three patients exhibited no clinical signs of CKD. Diabetes was present in a minority of the cohort, with only three patients diagnosed with the condition. In contrast, hypertension was recorded in five patients; all had an ACEI/ARB-based therapy; in addition, in three cases, it was combined with a diuretic, and a single one was taking a calcium-channel blocker (lercanidipine). The remaining six patients were normotensive (Table 2).

The study population consisted of patients aged 23 to 75 years, with a relatively uniform distribution across age groups. However, it should be noted that the entire cohort consisted of only 18 patients. Serum creatinine levels ranged from 63 to 125 µmol/L, while eGFR values spanned from 40 to over 122 mL/min/1.73 m^2^ (CKD-EPI 2021) for the females and from 60 to over 96 mL/min/1.73 m^2^ for the males, highlighting variability in kidney function among participants. The mean eGFR level for the first group that was sensitive to TRS treatment was 82.0 ± 16.4 µmol/L for the males, while the mean eGFR for the females was 67.7 ± 16.43 mL/min/1.73 m^2^ (Table 2). For the control group, we had similar results since the ranges of the creatinine were very close: 77.0 ± 13.4 µmol/L for the males, while the mean eGFR for the females was 69.3 ± 13.24 mL/min/1.73 m^2^.

The number of unsuccessful UTI treatments prior for all of the patients was 3.2 ± 0.75 (mean), whilst, for the main group, it was 3.4 ± 0.81 (mean). The range of unsuccessful treated UTIs caused by *K. pneumoniae* for all the patients varied from a minimum of 2 to a maximum of 5, illustrating the significant difficulty in finding suitable treatments. Most patients required at least three attempts before success was achieved. The average duration of symptomatic UTIs (with *K. pneumoniae*) for the main group of patients and their treatments prior to hospitalization in our clinic was 12.18 months (shortest period was 6 months, longest period was 18 months). Each patient had unsuccessful antibiotic treatment 7–30 days before the hospitalization in our clinic (Table 3).

From the documentation and data collected from our patients, we determined that eight antibiotics were prescribed and utilized during different periods and in varying sequences. All antibiotics were administered for at least a 5–7-day period, with some cases involving second-line therapy. Ceftriaxone was the most frequently prescribed antibiotic, administered to all 11 patients in the main group. This was followed by levofloxacin, amoxicillin/clavulanic acid, and imipenem, each prescribed to approximately 6–8 patients. Other antibiotics, including gentamicin, meropenem, and colistin, were used less frequently. The choice of treatment varied, often based on previous unsuccessful therapies for UTI and patient-specific factors, including allergies and the presence of antibiogram showing different resistance (Figure 1).

The same groups of antibiotics were used for treating the *K. pneumonia* UTI when the patients were admitted in our clinic. However, our choice of therapy was made after receiving the antibiogram. The first-line therapy lasted on average 8 days (the shortest period of intravenous treatment was 7 days, and the longest was 10 days). The choice of first-line treatment was based on the antibiogram provided by the microbiology laboratory (Table 4). All patients underwent a control microbiology test of their urine prior to discharge from the clinic. While all results were negative, similar outcomes had been observed following previous antibiotic treatments, and, shortly thereafter, patients experienced relapses and subsequent episodes of UTI caused by *K. pneumoniae*. As a result, all patients were placed on an extended second-line treatment with TRS, following the regimen outlined above.

They were on TRS for 10 weeks; every two weeks, the dosage was adjusted according to the regimen, and, every two to four weeks, they underwent microbiological urine tests, all of which came out negative. However, one patient experienced a delayed allergic reaction during treatment, occurring 3–4 weeks after the initiation of TRS therapy. She presented to the clinic with a skin rash, which resolved following discontinuation of the oral antibiotic and a short course of antihistamines. Despite the relatively shorter duration of second-line antibiotic treatment in her case, she also achieved a successful outcome, with a negative microbiology urine test and no symptoms related to UTI. Concurrently, the control group was observed, comprising patients exhibiting resistance to treatment with TRS according to the antibiogram of the urine analysis. It is noteworthy that, over the same period, only two patients in the cohort did not experience a new episode of UTIs. This finding indicates a significant disparity in the incidence of severe infections between the two groups over time.

## 4. Discussion

We have demonstrated that patients presenting with recurrent multidrug-resistant infections [9] *K. pneumonia* UTIs could be cured by using a preventive therapy with TRS following an initial IV antibiogram-adapted antibiotherapy.

The antibiotic combination of TRS is the preferred treatment for urinary tract infections (UTIs) due to its high efficacy against the uropathogens responsible for these conditions. The mechanism of action involves the inhibition of dihydrofolate reductase, leading to the blockade of tetrahydrofolic acid synthesis, a crucial intermediate in bacterial growth and reproduction. This antibiotic is suitable for various age groups, providing a versatile option in clinical practice [10].

In the treatment of acute uncomplicated UTIs, short courses of TRS lasting 3–5 days have been shown to yield excellent clinical outcomes, although this is not considered a standard guideline for managing uncomplicated urinary tract infections [11]. For cases involving frequent, recurrent, or chronic UTIs, long-term low-dose therapy is recommended. TRS is preferred for such treatment due to its low cost, favorable tolerability, and minimal risk of adverse effects [11].

Moreover, subinhibitory concentrations of TRS have demonstrated the ability to inhibit biofilm formation in select bacterial isolates. This is particularly significant for infections caused by resistant pathogens, such as Stenotrophomonas maltophilia [12], but not exclusively. Similar findings have been observed in patients treated for *K. pneumoniae* infections [13], Staphylococcus aureus, and for all three pathogens that TRS had effects at subtherapeutic concentrations [14]. Biofilms allow uropathogens to persist in a hostile environment, leading to chronic and recurrent UTIs. The extracellular matrix of biofilms protects bacteria from antibiotics and the immune system, complicating efforts to eradicate infections [15]. These uropathogenic strains harbor virulence genes that enhance biofilm formation, contributing to pathogenicity and making UTI treatment both difficult and prolonged, with a high potential for failure and recurrence. Understanding these factors is essential for developing targeted therapies that can improve treatment success rates [16]. Some studies suggest that factors influencing biofilm inhibition, such as the presence of TRS alongside environmental variables like temperature and pH, may play a role in the efficacy of treatment. These findings reinforce the continued use of TRS, particularly in settings where biofilm formation is prevalent [17]. Research conducted by Moon et al. has shown that subinhibitory concentrations of TRS effectively prevent biofilm formation, suggesting that, even at lower doses (400/80 mg per day), TRS can disrupt the early stages of biofilm development [12]. Additionally, various studies indicate that combining TRS with other antibiotic classes like quinolones enhances its effectiveness in treating biofilm-associated infections. For example, Río-Chacón et al. highlighted the importance of combining TRS with Levofloxacin, as these antibiotics were found to penetrate and disrupt established biofilms, reducing viable bacterial cells within the biofilm matrix [18]. Despite their effectiveness, fluoroquinolones, including levofloxacin, are associated with significant adverse effects, which necessitate careful consideration. For instance, fluoroquinolones have been linked to serious side effects, such as tendinopathy and tendon rupture, particularly in elderly patients and those concurrently receiving corticosteroid therapy [14]. These drugs are also known to cause peripheral neuropathy, which can be irreversible, as well as central nervous system disturbances, including confusion, dizziness, and an increased risk of seizures [14]. Furthermore, fluoroquinolone-induced dysglycemia, which can lead to both hypo- and hyperglycemia, poses particular risks to diabetic patients [14]. Recent studies have also linked fluoroquinolones to an increased risk of aortic aneurysm and dissection, leading to regulatory warnings and restrictions on their use, especially in individuals with predisposing cardiovascular conditions [14].

Given these risks, clinicians must carefully assess the benefits of combining TRS with fluoroquinolones, particularly in older or comorbid patients, to avoid severe adverse effects. Alternative therapies, such as the combination of TRS with colistin, have also demonstrated efficacy in treating carbapenem-resistant *Klebsiella pneumoniae*, providing a potential alternative when fluoroquinolones pose too great a risk [18]. Therefore, while combinations of TRS and fluoroquinolones can offer enhanced antimicrobial activity against biofilm-associated infections, their use must be carefully individualized to minimize adverse outcomes and ensure patient safety.

UTIs caused by resistant *K. pneumoniae* present a significant health challenge and are notoriously difficult to treat successfully over time. Treatment options are often limited because the isolates can exhibit resistance to multiple classes of anti-biotics. Multidrug-resistant (MDR) UTIs are becoming increasingly prevalent and represent a substantial public health threat due to the restricted therapeutic options available. In such cases, the synergistic effect of combining two different antibiotic classes is crucial for inhibiting and eradicating bacteria, ensuring a successful outcome. A study conducted by Su et al. explored the efficacy of combining TRS with colistin to treat carbapenem-resistant *K. pneumoniae* infections. Their findings showed that the combination of TRS and colistin had strong synergistic effects, resulting in rapid bacterial clearance and a significant improvement in treatment success, particularly against strains highly resistant to other antibiotics [19]. While TRS alone has demonstrated synergistic potential, its combination with other antibiotic classes, such as colistin, can further enhance its effectiveness. TRS may target the folate biosynthetic pathway to prevent biofilm formation by *K. pneumoniae*, a mechanism previously reported for resistant uroinfections caused by *E. coli* and *A. baumannii* [12].

It is crucial to recognize that colistin can be nephrotoxic, with a dose-dependent relationship to the development of acute kidney injury (AKI) [20]. This toxicity is primarily attributed to its direct effects on renal tubular epithelial cells, which pose a significant risk, especially in patients with preexisting renal impairment. Therefore, the close monitoring of renal function throughout treatment is essential [21].

Although less common, colistin-induced neurotoxicity can manifest as dizziness, confusion, neuromuscular blockade, and, in severe cases, respiratory failure due to paralysis [20].

Additionally, colistin therapy presents pharmacokinetic challenges, including a narrow therapeutic window and variability in drug absorption and metabolism, which can lead to suboptimal dosing [19,20,21]. This, in turn, increases the risk of resistance development and exacerbates concerns about toxicity. Given these risks, clinicians must carefully weigh the benefits of TRS and colistin combination therapy against the potential for severe adverse events, particularly in vulnerable patient populations.

*Klebsiella pneumoniae* is a common cause of urinary tract infections, particularly in healthcare settings, and is especially prevalent in hospital-acquired infections. The prevalence of *K. pneumoniae* in UTIs varies globally and depends on factors such as regional differences in antimicrobial resistance patterns. For instance, a study in Burkina Faso found that Klebsiella spp. accounted for 7.18% of UTI cases [22]. However, other studies have reported varying prevalence rates depending on geographical location and hospital settings. A study conducted in Hungary analyzed antibiotic prescription patterns for hospitalized UTI patients and identified a significant rate of misdiagnosis, which can influence the reported prevalence rates of *K. pneumoniae* infections [23]. Similarly, surveillance data from Poland revealed that *Klebsiella* spp. was the second most frequently isolated pathogen in UTIs, following *E. coli*, emphasizing the need for region-specific empirical treatment guidelines [24]. Analyzing cumulative antibiograms from a European medical center highlighted the necessity of tailored empirical therapy based on local resistance patterns. This study found that *K. pneumoniae* exhibited variable susceptibility to first-line antibiotics, reinforcing the need for continuous epidemiological surveillance to guide appropriate treatment choices [23]. These findings underscore the regional variability in *K. pneumoniae* prevalence in UTIs and the importance of monitoring antimicrobial resistance trends to optimize treatment strategies.

A conservative estimate based on 2019 global data suggests that *K. pneumoniae* may have been responsible for approximately 28.3 million UTI cases worldwide that year [25]. Research on antimicrobial resistance has highlighted *K. pneumoniae* as a significant causative agent of UTIs, especially in cases of multidrug resistance, where it accounts for a comparable proportion of infections for the hospitalized patients [26]. TRS has been shown to be effective in preventing recurrent UTIs, with recurrence rates reduced by over 50% in women receiving low-dose prophylaxis [25], for example, 400/80 mg per day. When used prophylactically, TRS has been found to reduce mortality and the incidence of opportunistic infections compared to other antibiotics used in immunocompromised patients [27].

Several reports have highlighted complications associated with long-term TRS therapy in patients, including the following:-Hematological Toxicity: TRS can inhibit folate metabolism, leading to complications such as pancytopenia, megaloblastic anemia, and thrombocytopenia. Patients with pre-existing folate deficiency are at increased risk for these complications [28].-Renal Effects: Hyperkalemia is a common adverse effect due to impaired renal potassium excretion by trimethoprim [29]. Additionally, sulfamethoxazole-induced crystalluria can result in kidney stones and hematuria [30].-Dermatological Reactions: Severe reactions, such as Stevens–Johnson syndrome (SJS) and toxic epidermal necrolysis (TEN), have been associated with prolonged TRS use [31].-Hepatotoxicity: Liver injury, including transaminitis and cholestasis, has been documented, with rare cases of fulminant liver failure in patients on long-term TRS therapy [32].-There is a potential risk to the fetus during pregnancy if TRS is prescribed. TRS acts as a folic acid antagonist, a nutrient essential for fetal development, particularly during the early stages of pregnancy. The inhibition of folic acid can increase the risk of neural tube defects and other congenital abnormalities [33], as well as elevate the risk of miscarriage [34].

Although these side effects are rare and occur in a small proportion of patients relative to the widespread use of TRS, healthcare providers should remain vigilant, especially when prescribing the drug for prolonged periods. Additionally, treatment regimens should be sufficiently long to ensure effectiveness, as resistance to TRS has been reported. The emergence of resistant strains, particularly in *E. coli*, can reduce the efficacy of TRS, necessitating more aggressive treatment strategies [35]. Our study clearly demonstrates that TRS can be highly effective for recurrent urinary infections, even in patients who have previously undergone at least two unsuccessful treatments. Notably, we observed no significant adverse effects, except for a late allergic reaction in one patient, who ultimately achieved a successful outcome.

Furthermore, TRS is economically viable and widely used for a variety of therapeutic indications [36]. Despite the range of alternative antibiotics available, achieving successful outcomes in UTI treatment requires adherence to certain principles by healthcare professionals, with patient safety being the utmost priority [37]. Our team strongly supports the therapeutic viability of TRS as a treatment option for recurrent urinary infections, as evidenced by our study, while emphasizing the need for appropriate dosing (considering factors such as body mass, age, gender, renal function, and comorbidities) to minimize the risk of complications.

### Strengths and Limitation

This study provides valuable insights into the efficacy of a structured, long-term TRS regimen for treating recurrent multidrug-resistant *K. pneumoniae* urinary tract infections (UTIs). One of the key strengths of this research is its clinical relevance, as it addresses a critical gap in the management of MDR *K. pneumoniae* UTIs, a condition with limited therapeutic options. The structured TRS regimen offers a potentially effective and cost-efficient second-line treatment. Moreover, this study demonstrates strong microbiological eradication, with all patients achieving negative urine cultures following the treatment regimen, highlighting the high efficacy of the treatment protocol. Additionally, the stepwise reduction in TRS dosage provided a systematic approach to long-term treatment, which may help minimize resistance development while maintaining therapeutic efficacy and reducing the risk of side effects associated with prolonged antibiotic use.

However, this study also has several limitations. The small cohort size, with only 18 patients, limits the generalizability of the findings, necessitating larger, multicenter studies to confirm the effectiveness of the structured TRS regimen. Also, the control group was followed only as an outcome for the treatment, and, in the future, this group needs a larger number of patients as well as detailed information, like types of antibiotic used in previous treatments of the recurrent UTIs. In this way, we can compare TRS efficacy with other second-line therapies. Finally, as this research was conducted in a single clinical setting, its applicability may be constrained in broader patient populations with varying resistance profiles.

## 5. Conclusions

This study demonstrates that TRS can serve as an effective and cost-efficient second-line therapy for patients with recurrent multidrug-resistant *K. pneumoniae* urinary tract infections (UTIs). A structured, long-term TRS regimen led to complete microbiological eradication and sustained symptomatic relief in all patients, despite their history of multiple treatment failures with other antibiotics. In the context of the increasing prevalence of antimicrobial resistance, the affordability, accessibility, and manageable side-effect profile of TRS position it as a viable therapeutic option, particularly in settings with limited access to novel antimicrobial agents. Additionally, the potential of TRS to inhibit biofilm formation further underscores its relevance in managing persistent infections.

In conclusion, TRS therapy offers a valuable treatment alternative for recurrent MDR *K. pneumoniae* UTIs, providing a practical option within antimicrobial stewardship strategies aimed at preserving antibiotic efficacy while ensuring favorable patient outcomes.

## Figures and Tables

**Figure 1 microorganisms-13-00422-f001:**
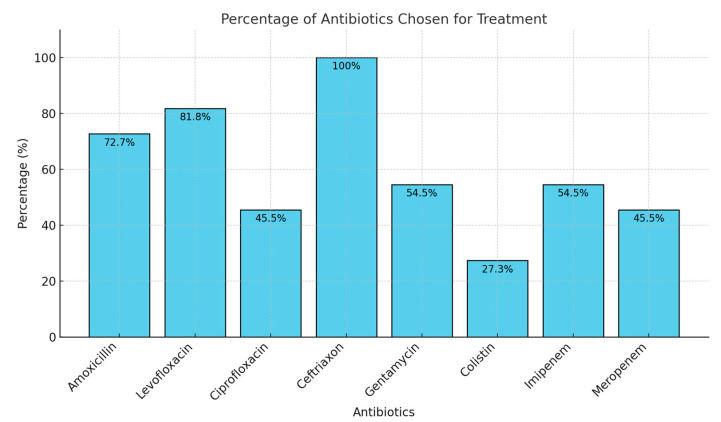
Type of antibiotics used in the main group of patients to treat *K. pneumonia* urinary tract infections before they were admitted in our clinic.

**Table 1 microorganisms-13-00422-t001:** Demographic information and comorbidities of patients with TRS treatment.

Patient	Sex	Age (years)	Serum Creatinine (μmol/L)	CKD Stage	Diabetes	Hypertension
**1**	Male	51	100	II	Yes	No
**2**	Female	75	109	IIIa	No	Yes
**3**	Male	45	90	No	No	No
**4**	Male	65	76	I	No	No
**5**	Female	60	103	IIIa	No	Yes
**6**	Female	59	91	II	No	No
**7**	Female	63	130	IIIb	Yes	Yes
**8**	Male	51	125	II	No	Yes
**9**	Female	23	63	No	No	No
**10**	Female	27	68	No	No	No
**11**	Female	71	123	IIIb	Yes	Yes

Abbreviations: CKD, Chronic kidney disease.

**Table 2 microorganisms-13-00422-t002:** Kidney function of the patients sensitive to TRS treatment.

Characteristic Statistics	
	N = 11
**eGFR (mL/min/1.73 m^2^)**	
**Males**	
*n*	4
Mean (SD)	82.0 (16.43)
Minimum	60.0
25th Percentile (1st Quartile)	69.5
Median	86.0
75th Percentile (3rd Quartile)	94.5
Maximum	96.0
**Females**	
*n*	7
Mean (SD)	67.7 (33.50)
Minimum	40.0
25th Percentile (1st Quartile)	41.0
Median	54.0
75th Percentile (3rd Quartile)	108.0
Maximum	122.0
**Total**	
*n*	11
Mean (SD)	72.9 (28.40)
Minimum	40.0
25th Percentile (1st Quartile)	46.0
Median	63.0
75th Percentile (3rd Quartile)	96.0
Maximum	122.0

Abbreviation: SD, standard deviation; eGFR, estimated glomerular filtration rate.

**Table 3 microorganisms-13-00422-t003:** First choice for antibiotic treatment in our clinic for all the patients.

Characteristic statistics	N = 11
**Duration of treatment (months)**	
*n*	11
Mean (SD)	12.2 (3.40)
Minimum	6.0
25th Percentile (1st Quartile)	10.0
Median	12.0
75th Percentile (3rd Quartile)	15.0
Maximum	18.0
**Choice of antibiotic**	
*n* (%)	
Amoxicillin/Clavulanic acid	1 (9.1%)
Colistin	3 (27.3%)
Imipenem	4 (36.4%)
Meropenem	2 (18.2%)
Piperacillin-Tazobactam	1 (9.1%)
Total	11 (100%)

Abbreviation: SD, standard deviation.

**Table 4 microorganisms-13-00422-t004:** Microbiology results of urine samples.

Characteristic statistics	N = 11
**All patients had a result for *Klebsiella pneumoniae* 10^5^**	
**Patient Number**	**Anitbiotics to which the bacteria was “Sensitive”**
Patient No.1	**Amoxicillin/Clavulanic acid**; Imipenem; Meropenem; Gentamycin; TRS;
Patient No.2	Imipenem; **Meropenem**; Gentamycin; TRS
Patient No.3	**Piperacilin/Tazobactam**; Imipenem; Meropenem; Colistin; TRS; Fosfomycin;
Patient No.4	**Colistin**; TRS; Nitrofurantoin;
Patient No.5	Ceftriaxon; **Imipenem**; Gentamycin; TRS;
Patient No.6	**Meropenem**; Gentamycin; TRS; Nitrofurantoin;
Patient No.7	**Imipenem**; Amikacin; Gentamycin; TRS;
Patient No.8	**Imipenem**; TRS; Fosfomycin;
Patient No.9	**Colistin**; TRS; Fosfomycin;
Patient No.10	**Imipenem**; Amikacin; Gentamycin; TRS;
Patient No.11	**Colistin**; TRS;

Additional explanation: The antibiotic in bold is the first-line treatment choice for each patient.

## Data Availability

Data are unavailable due to privacy or ethical restrictions.

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
