# Peer review of "Urinary Tract Infections Caused by Klebsiella pneumoniae and Prolonged Treatment with Trimethoprim/Sulfamethoxazole"

_microorganisms, 2025, doi:10.3390/microorganisms13020422_

Round 1

Reviewer 1 Report

Comments and Suggestions for Authors

'Urinary Tract Infections with Klebsiella pneumonia and Long 2 Treatment with Trimethoprim/Sulfamethoxazole' is a clearly written manuscript which is easy to understand for the readers.

However the patients showing positive response to the treatment of TMP-SMX is still small, just 11. The question remains: 11 out of how many? How were the 11 patients shortlisted? Are these the only patients during the course of study who visited the clinic showing Antibiotic resistance while treatment for UTIs? It can be easily understood that many resistant patients must have been selected, out of which only 11 specific cases, with positive expected results are reported. Please detail the initial screening process, the duration and dates when this was performed and how the 11 patients were zeroed in. What is the negative data in this case, any patients who also started with similar resistance and infection, but could not be healed with TMP-SMX? This information is critical, for the benefit of numerous patients. 

The introduction must go into details of various treatment options against K.pneumonae and the nuisance of antibiotic resistance. Afterall, a widespread use of TMP-SMX may also lead to resistance.  Moreover, recent related literature needs to be cited as there are very few coming from last few years. Following can help to improve the introduction:

https://doi.org/10.1093/ofid/ofy351

https://doi.org/10.1002/jcla.24619

https://doi.org/10.1080/07391102.2020.1767209

Do the authors propose that TMP-SMX can be prescribed at initial stages for UTI by Klebsiella pneumonia?

Minor typo errors in keywords

Author Response

Reviewer â„–1

'Urinary Tract Infections with Klebsiella pneumonia and Long 2 Treatment with Trimethoprim/Sulfamethoxazole' is a clearly written manuscript which is easy to understand for the readers.

However the patients showing positive response to the treatment of TMP-SMX is still small, just 11. The question remains: 11 out of how many? How were the 11 patients shortlisted? Are these the only patients during the course of study who visited the clinic showing Antibiotic resistance while treatment for UTIs? It can be easily understood that many resistant patients must have been selected, out of which only 11 specific cases, with positive expected results are reported. Please detail the initial screening process, the duration and dates when this was performed and how the 11 patients were zeroed in. What is the negative data in this case, any patients who also started with similar resistance and infection, but could not be healed with TMP-SMX? This information is critical, for the benefit of numerous patients.

The introduction must go into details of various treatment options against K.pneumonae and the nuisance of antibiotic resistance. Afterall, a widespread use of TMP-SMX may also lead to resistance.  Moreover, recent related literature needs to be cited as there are very few coming from last few years. Following can help to improve the introduction:

https://doi.org/10.1093/ofid/ofy351

https://doi.org/10.1002/jcla.24619

https://doi.org/10.1080/07391102.2020.1767209

Do the authors propose that TMP-SMX can be prescribed at initial stages for UTI by Klebsiella pneumonia?

Minor typo errors in keywords

Answer:

We would like to thank the reviewer for the feedback and we would like to answer the stated comments point by point:

  1. Indeed, it is a small group of patients that have fallen into the same profile – having at least two unsuccessful treatments of Klebsiella pneumonia over 6-18 months, which had a urine microbiology performed in Saint Anna Hospital and in the end have agreed to have a treatment with TMP-SMX over long period of time. All these patients had to undergo the same criteria:
  • Previous results with Klebsiella pneumonia with at least two antibiotic treatments with no success over time from the antibiotic course
  • Having ultrasound prove for lower tract UTI – thickening of the bladder wall over 5 mm, post-void residual volume, echogenic debris in the bladder and/or bladder mucosal irregularity.
  • In between the time of the separate antibiotic treatments had still symptoms. Patients who had negative microbiology for example over 6 months were not included.
  • The urine microbiology had to have “Sensitive” result for TMP-SMX
  • Had no allergies to antibiotics/other medication, because of the risk reaction to the antibiotic (had one case of late reaction although the patient had no medical data for allergies)
  • Have agreed to be followed over time.
  • All the patients were over 18 years old.
  1. The selection of patients was based on the following criteria: firstly, the necessity to conduct a follow-up of Klebsiella pneumoniae cases; secondly, the requirement to identify cases of patients who had been admitted to the clinic with such an infection and had been treated multiple times, with a particular focus on elderly patients.
  2. All 11 patients are followed еven in this moment, because we want to see if we have success for long time, so we can get more data and try the long treatment for more patients. The last patient has finished the antibiotic treatment almost 6 months ago and the first one has finished it 11 months ago and until this moment we have no new episode of UTI.
  3. There were 7 patients who were resistant to TMP-SMX, so they were not added in the group. They were also followed up and they do not have such a successful outcome over the past few months. We sincerely apologies for not adding the second group – patients with positive urine analtysis for K. pneumonia, but resistent to TRS treatment. Now the information is added in the text.

Materials and methods

The cohort was divided into two groups: the first group comprised 11 patients (seven females and four males), and microbiological analysis revealed that all of them exhibited a positive response to treatment with TRS. The second group, comprising seven patients (five females and two males), served as the control group. All patients in the control group exhibited positive results from microbiology urine analysis for K. pneumoniae, yet all demonstrated resistance to TRS treatment according to the antibiogram.   

Results:

Concurrently, the control group was observed, comprising patients exhibiting resistance to treatment with TRS according to the antibiogram of the urine analysis. It is noteworthy that over the same period, only two patients in the cohort did not experience a new episode of UTIs. This finding indicates a significant disparity in the incidence of severe infections between the two groups over time.

  1. We would like to thank deeply to Reviewer â„–1 for the suggested papers that have helped us to improve greatly the Introduction of the paper and more information was added for K. pneumonia and the nuisance of antibiotic resistance.
  2. The last question was if we propose that TMP-SMX can be prescribed at initial stages for UTI by Klebsiella pneumonia: As stated in the introduction, in the last sentence, we aim to demonstrate whether TMP-SMX can be used as a long-term, effective and cost-effective second-line therapy for patients with recurrent MDR K. pneumoniae infections.

The corrected version is uploaded for Reviewer â„–1. 

Reviewer 2 Report

Comments and Suggestions for Authors

This manuscript requires careful revision.

Lines 36-38. A reference is needed to support the assertion contained here.

Lines 39-41. This passage should be rephrased as it contains an internal contradiction; it now reads as trimethoprim-sulfamethoxazole and ciprofloxacin have long been discontinued despite the fact that high rates of resistance to them have been documented. The cited publication explicitly states that high rates of resistance to trimethoprim-sulfamethoxazole and ciprofloxacin preclude their use as empirical treatment for UTI.

Lines 43-46. In the cited publication [2] (which is a collection of clinical guidelines), data on resistance to trimethoprim-sulfamethoxazole and other antibiotics pertain mainly to E. coli, but not to Klebsiella. The reviewer recommends using another publication for citation here.

Lines 48-51. The same reviewer's comment on the need to correctly use publication citations. The content of [2] is out of context in the manuscript text in lines 48-51.

Materials and Methods Section.

This section needs extensive revision; reading the section raises many questions, which the reviewer has written below.

Why was this study design chosen? Are there only 11 patients in the study enough? Why is there no control group?

What were the criteria for selecting patients for this study? Was the study registered/approved by the Ethics Committee? This is not written in the manuscript.

How was the microbiological study performed? How was Klebsiella pneumoniae isolated and identified? How was antibiotic susceptibility determined?

On what basis was trimethoprim-sulfamethoxazole prescribed after successful use of antibiotics (as the authors say in lines 67-68)?

What determined the duration of treatment with trimethoprim-sulfamethoxazole?

Why was a one-month interval chosen for urine analysis?

Results section. This section is written in a chaotic manner and does not contain data that would support the authors' assertion about the effectiveness of their proposed treatment regimen; the section requires careful revision. Given the title of the manuscript (Urinary tract infections with Klebsiella pneumonia and long treatment with Trimethoprim/sulfamethoxazole) and the authors' stated objective, the reviewer recommends paying more attention to the description of microbiological results; this section contains the minimum possible number of them.

The section contains a significant number of demographic and clinical indicators; some of them are redundant, since they do not reflect the problems that the authors had to solve to achieve the objective of their study.

The large number of tables in the text complicates its understanding; at the same time, the description of the results itself is contradictory, for example, in lines 94-95 the authors write about an insignificant (as the authors say) excess of the proportion of women among patients. But in fact, women account for almost two thirds.

The data in Table 1 and Table 2 are partially duplicated; both contain demographic data, but in one case (Table 1) these data are for each patient, while the other table contains processed data in the form of means, medians, etc. Why do we need Table 1 with raw demographic data if we have Table 2?

In lines 117-118, the authors write that the age distribution is uniform; the reviewer notes that the minimum value in Table 2 is 23 years, and the maximum is 75 years. Is it correct to say this, given that there are only 11 patients in the study?

The text of the manuscript does not contain data that would demonstrate the efficacy and safety of therapy using trimethoprim-sulfamethoxazole. The Materials and Methods indicate that urine was analyzed during therapy with trimethoprim-sulfamethoxazole; this is not shown in the Results section.

And in general, was the primary therapy successful? This question is not answered in the section. And again, the question arises about the necessity of continuation of therapy with trimethoprim-sulfamethoxazole.

The Discussion section needs to be revised, since in its current form it resembles a literature review that is not related to the authors' results. The authors' results are not discussed here at all.

The conclusions in the manuscript are not based on the results described by the authors and are declarative in nature.

Author Response

Dear Reviewer â„–2, 

Our reply is uploaded as a file. 

Thank you in advance!

Reviewer 3 Report

Comments and Suggestions for Authors

The manuscript presents an interesting analysis of several cases of UTIs (or recurrent UTIs? – this needs clarification) treated with prolonged therapy using trimethoprim and sulfamethoxazole. However, to make the text suitable for publication in a scientific journal, it requires a thorough rewrite. It also needs comprehensive proofreading and editing by a professional scientific translator, preferably a native English speaker.

Errors are evident as early as in the title, where, in addition to the unusual use of the preposition “with,” the bacterial species name is incorrectly written. I suggest changing the title to “Urinary Tract Infections Caused by Klebsiella pneumoniae and Prolonged Treatment with Trimethoprim/Sulfamethoxazole.”

Throughout the text, bacterial species names should be italicized. Additionally, the principle should be followed that the full name of the bacterium (e.g., Klebsiella pneumoniae) is used upon first mention, and subsequently, the genus name is abbreviated (e.g., K. pneumoniae).

Instead of the abbreviation “TMP-SMX” (likely inspired by commercial laboratory materials), I propose using the abbreviation “TRS,” in line with the EUCAST system for antimicrobial abbreviations (https://www.eucast.org/fileadmin/src/media/PDFs/EUCAST_files/Disk_test_documents/Disk_abbreviations/EUCAST_system_for_antimicrobial_abbreviations.pdf).

Keywords contain typos!!!

Lines 32–38: Sources are missing. Every piece of information provided in the Introduction must cite a reference. Speculative content belongs in the Discussion. Please address missing references throughout the entire Introduction!

Line 43: Which guidelines recommend fluoroquinolones as the first-line treatment for UTIs Please update the Introduction based on the EUA treatment guidelines - https://uroweb.org/guidelines/urological-infections/chapter/the-guideline.

Materials and Methods need to be expanded. Crucially, information is missing on how samples for microbiological studies were collected and processed.

Lines 59–59: What does “at least two unsuccessful antibiotic treatments” mean? Over what timeframe? For which infections—only UTIs?

Line 64: The superscript for the exponent is missing.

Lines 64–65: What UTI symptoms did the patients exhibit? Were these infections of the lower or upper urinary tract? Were these recurrent UTIs? What criteria were used?

Lines 71–83: This section is not very clear. I suggest illustrating it with a graphic comparing antibiotic administration regimens and the timing of sample collection for laboratory testing.

Results are described in reasonable detail. However, microbiological results for the patients are missing. It is crucial to present the antibiograms of the Klebsiella pneumoniae isolates.

Line 98: What criteria were adopted for “multi-drug-resistant”? Please cite the source.

Table 2: Unnecessary duplication of many details from Table 1. The patient group is so small that basic statistics do not need to be calculated for the reader.

Line 126: Unsuccessful treatment of what? UTI?

Table 4: “Average number of treatments” – over a lifetime? Only for UTIs? Specify!

Table 4: Data on antibiotics is unnecessary as it duplicates Figure 1 (which is clearer than the table).

Lines 135–144: Antibiotic names should not be capitalized.

Figure 1: Add percentage values.

Information is missing on which antibiotics were used as subsequent lines of therapy for individual patients.

Lines 149–150: Did you wait for antibiogram results before starting antibiotic therapy for UTIs? Is UTI not treated empirically in your facility? Please explain and cite relevant recommendations.

Table 5: I assume that since all patients received trimethoprim with sulfamethoxazole, all Klebsiella isolates were susceptible to it. Why, then, were colistin or carbapenems administered if the isolates were susceptible to trimethoprim/sulfamethoxazole? Please explain and cite relevant recommendations.

Line 156: 10 weeks or 2 weeks? On page 2, you wrote that trimethoprim/sulfamethoxazole was administered for 2 weeks.

Line 162: “Achieved a successful outcome” – what does this mean?

Line 166: Why was the therapy intravenous (IV) and not oral (PO)? Please explain and cite relevant recommendations.

Lines 173–178: The article by Al-Badr et al. is a review, not a set of recommendations. There is a significant difference between reviews and official guidelines.

Lines 197–202: Please discuss this in the context of the adverse effects of fluoroquinolones.

Lines 209–212: Please discuss this in the context of the adverse effects of colistin.

Line 219: The major pathogen in UTIs in general is E. coliKlebsiella is one of the three most common pathogens in HA-UTIs (hospital-acquired), along with E. coli and Enterococcus.

Line 222: Cite additional sources—one study from Burkina Faso is not sufficient. Suggested references:

 https://www.mdpi.com/2079-6382/14/1/14

 https://www.mdpi.com/2077-0383/12/19/6270

 https://www.mdpi.com/2079-6382/12/12/1689

Lines 233–246: Add information on the risks for the mother and fetus during pregnancy.

Strengths and Limitations section is missing—please include it!

Why is there no bioethics committee approval? Why was it not required? Do local regulations in Bulgaria allow this?

Comments on the Quality of English Language

a need of comprehensive proofreading and editing by a professional scientific translator, preferably a native English speaker

Author Response

Dear Reviewer â„–3, 

Our reply is uploaded as a file. 

Thank you in advance!

Reviewer 4 Report

Comments and Suggestions for Authors

Scientific names should be in italics. The whole manuscript needs to be revised.

References must be formatted according to the journal’s guidelines.

The study’s aims must be mentioned in the Abstract (Background section). The applied methodologies must be specified and the main results (values) must also be highlighted.

Lines 33-38: References are missing.

The introductory section is not adequate. Much more is expected from the introductory section of a manuscript that will be submitted to an international journal. The authors should deeply analyze the state of the art and justify the need to conduct their research.

The study’s objectives shouldn’t be in the Materials and Methods section.

Line 58: Why 11 patients? What about the sample’s representativeness?

Where did you recruit the patients? What are their characteristics?

When discussing your results, elaborate on your study’s limitations.

Author Response

Dear Reviewer â„–4, 

Our reply is uploaded as a file. 

Thank you in advance!

Round 2

Reviewer 1 Report

Comments and Suggestions for Authors

The authors have sufficiently improved the manuscript. I have no further comments.

Author Response

We are extremely grateful to Reviewer #1 for the opportunity to improve our work. 

Reviewer 2 Report

Comments and Suggestions for Authors

The reviewer is completely satisfied with the authors' responses; the quality of the manuscript has improved after revision.

As a recommendation, the reviewer suggests changing the structure of subsection 2.1 (Study Design). The reviewer believes that the text in lines 107-121 should precede the text in lines 94-106. At least then the logic of the presentation is observed. The reviewer would like to note that his remark is not critical and leaves this decision to the discretion of the authors.

Author Response

We are extremely grateful to Reviewer â„–2 for the opportunity to improve our work. 

 We will be utilising the recommendation for further improvement of the paper. 

Reviewer 3 Report

Comments and Suggestions for Authors

After making corrections, the article sounds scientific and is suitable for publication after some minor corrections. I would like to thank the authors for their detailed responses to my comments – my doubts have been dispelled.

Minor corrections:

Line 23 – italicize “K. pneumoniae”, please check the entire article for this

Line 69 – use “Enterobacterales” instead of “Enterobacteriacae”

Line 116 – use “susceptibility” instead of “sensitivity” - please check the entire article for this; in this context we are talking about antibiotic sensitivity, not about the feelings of bacteria ;)

Line 243-245 – please correct the citation of tables, something is wrong here

Table 4 – typo in “pneumoniae”; typo in “Clavulonic”; Monoral?; slash instead of dash in “Piperacillin/Tazobactam” Linezolid??? (for gram-negative!)

Please explain why you chose these substances for targeted treatment. In many cases, the bacteria were sensitive to aminoglycoside, why didn't you choose it?

Author Response

We would like to express our sincere gratitude to Reviewer â„–3 for the opportunity to refine our work and for the time given for reviewing our work.

Again we would like to answer to the comments point by point: 

Minor corrections:

Line 23 – italicize “K. pneumoniae”, please check the entire article for this

Thank you for the comment, we have missed that one and it is corrected. 

Line 69 – use “Enterobacterales” instead of “Enterobacteriacae”

Thank you for pointing that, we have corrected it. 

Line 116 – use “susceptibility” instead of “sensitivity” - please check the entire article for this; in this context we are talking about antibiotic sensitivity, not about the feelings of bacteria ;)

Well spoted! Those poor bacteria must be relieved to know we're not questioning their emotional well-being. 

We have checked the entire article and ensure we're talking about their antibiotic susceptibility, not their delicate feelings.

Thank you for the insightful remark! 

Line 243-245 – please correct the citation of tables, something is wrong here

Please excuse us for the technical mistake. Two tables were deleted and one was added so this is Table 4, not Table 5. 

Table 4 – typo in “pneumoniae”; typo in “Clavulonic”; Monoral?; slash instead of dash in “Piperacillin/Tazobactam” Linezolid??? (for gram-negative!)

Thank you for improving the table - indeed there typos and they are corrected now. 

Linezolid is technical mistake by the team, we have double checked the antibiogram from microbiology. It is indeed written Linezolid to the results, but with a blind spot, without a result to it and for some reason was written in the table. 

Our sincerest apologies! 

Please explain why you chose these substances for targeted treatment. In many cases, the bacteria were sensitive to aminoglycoside, why didn't you choose it?

Five antibiograms were obtained, with a possible choice of Gentamicin treatment. One patient had a history of allergies to aminoglycosides, while one patient had CKD 3b (eGFR < 40 ml/min/1.73m2). Due to this, it was decided to use Imipenem with adequate dose adjustment. In the other three cases, previous Gentamicin treatment had been unsuccessful.

These factors guided the decision to pursue an alternative antibiotic treatment regimen. 

Corrected version and clean version will be uploaded!

Reviewer 4 Report

Comments and Suggestions for Authors

Thank you for providing adequate answers to all my concerns.

Author Response

We would like to express our gratitude to Reviewer No. 4 for the opportunity to refine our work and for the possibility of publication in the journal.